# Perspectives on Nickel Hydroxide Electrodes Suitable for Rechargeable Batteries: Electrolytic vs. Chemical Synthesis Routes

**DOI:** 10.3390/nano10091878

**Published:** 2020-09-19

**Authors:** Baladev Ash, Venkata Swamy Nalajala, Ashok Kumar Popuri, Tondepu Subbaiah, Manickam Minakshi

**Affiliations:** 1Department of Chemistry, M.P.C. (Autonomous) College, Baripada, Odisha 757003, India; baladevash@gmail.com; 2Department of Chemical Engineering, VFSTR (Vignan’s Foundation for Science, Technology and Research), Vadlamudi 522 213, India; venkataswamy.nalajala@gmail.com (V.S.N.); drpak_chem@vignan.ac.in (A.K.P.); 3College of Science, Health, Engineering & Education, Murdoch University, Perth, WA 6150, Australia

**Keywords:** battery grade, nickel hydroxide, battery, energy storage, electrochemical

## Abstract

A significant amount of work on electrochemical energy storage focuses mainly on current lithium-ion systems with the key markets being portable and transportation applications. There is a great demand for storing higher capacity (mAh/g) and energy density (Wh/kg) of the electrode material for electronic and vehicle applications. However, for stationary applications, where weight is not as critical, nickel-metal hydride (Mi-MH) technologies can be considered with tolerance to deep discharge conditions. Nickel hydroxide has gained importance as it is used as the positive electrode in nickel-metal hydride and other rechargeable batteries such as Ni-Fe and Ni-Cd systems. Nickel hydroxide is manufactured industrially by chemical methods under controlled conditions. However, the electrochemical route is relatively better than the chemical counterpart. In the electrochemical route, a well-regulated OH^−^ is generated at the cathode forming nickel hydroxide (Ni(OH)_2_) through controlling and optimizing the current density. It produces nickel hydroxide of better purity with an appropriate particle size, well-oriented morphology, structure, et cetera, and this approach is found to be environmentally friendly. The structures of the nickel hydroxide and its production technologies are presented. The mechanisms of product formation in both chemical and electrochemical preparation of nickel hydroxide have been presented along with the feasibility of producing pure nickel hydroxide in this review. An advanced Ni(OH)_2_-polymer embedded electrode has been reported in the literature but may not be suitable for scalable electrochemical methods. To the best of our knowledge, no such insights on the Ni(OH)_2_ synthesis route for battery applications has been presented in the literature.

## 1. Introduction

Electrical energy storage devices, comprising batteries and capacitors, employ both aqueous and non-aqueous electrolytes in either liquid or solid state. To mitigate safety concerns, aqueous (aq.) electrolytes are the natural choice in this field [1,2]. Water is much cheaper than organic solvents (such as ethylene carbonate, propylene carbonate, etc.), is easier to purify, and has fewer recycling/disposal issues. The main disadvantage of using an aqueous electrolyte is that the battery voltage is limited to ~2 V to avoid electrolysis. To compensate for the effect of low voltage, the storage capacity and the charge-discharge characteristics of the aqueous system must be significantly higher to those of the non-aqueous systems, in which liquid electrolytes consist of a lithium salt (1 M LiClO_4_) in an organic solvent. Two important quantities for electrical energy storage devices are specific energy and cycle life, and typical values are shown in Table 1. The four main systems of rechargeable (secondary) batteries are compared in this table. The Ni-Cd, Ni-MH, and Li-ion are receiving greater attention due to their practical applications. Based on reliability, safety, and cost, aqueous-based Ni-Cd and Ni-MH systems have shown great progress in electronic devices, defense equipment, and space research markets. The deterioration and irreversible effects of cadmium electrode (related to changes in crystal sizes, electrolyte redistribution) limits its applications in space and other stationary performance. The availability of stable hydrogen storage alloys as the negative electrode material provided the impetus for the creation of the latter type, nickel metal hydride (Ni-MH) batteries. The hydrogen storage alloy involves an “insertion of H^+^” acting as negative electrode material that has replaced the well-known cadmium (Cd), which is environmentally harmful in the nickel-cadmium (Ni-Cd) system. However, in the Ni-MH version, the positive electrode and the potassium hydroxide (KOH) electrolyte remained the same. Due to the nature of the electrolyte, this battery is classified as an alkaline storage rechargeable (secondary) battery. Ni-MH quickly replaced the Ni-Cd for electronic applications due to its enhanced voltage and storage capability with improved efficiency. This is an environmentally friendly battery as there are no toxic components involved as seen in the competitive lead acid battery.

In developed Ni-MH batteries, the positive electrode is nickel hydroxide (NiOOH) used with optimum amounts of additives (such as Co(OH)_2_, Y_2_O_3_, graphite powders, etc.) to enhance the electrical conductivity of the cathode for higher charge efficiency [6,7]. Increasing the amount of additives during the electrochemical synthesis of the NiOOH electrode resulted in active material deposition of reduced mechanical strain. This approach results in enhanced cycle life [8]. In the last few years, the technology of nickel hydroxide production and its utilization techniques in Ni-Cd and Ni-MH batteries has increased. Because of the importance of secondary batteries and major investments have been made in this area in the last decade; a perspective on these materials including mass scale production could be timely to the battery community. Ni-MH batteries have been drawing increasing attention due to their high rate discharge with relatively less heat dissipation [9,10]. This technology has gained acceptance throughout the world market for their ever-increasing demands for communication, space, and defense applications because of its unique features such as long-term cycle life at deep depth of discharge (DoD).

In this perspective, an attempt is made to summarize the information on Ni-MH battery technology with an emphasis to synthesize routes and bring insights into the chemistry involved and product formation mechanism in nickel hydroxide. Higher utilization of the NiOOH active material appears to be achieved with the electrochemical synthesized route compared to the chemical route. Electrochemical impregnation of the cathodic reduction of nitrate ion within the pores of the porous nickel plaque cathode provides very uniform loading of the NiOOH electrode [9,10,11]. For the chemical impregnation method, corrosion rate is high and that results in void volume. The difference in mechanism is discussed in this perspective.

## 2. Description of Basic Materials of Nickel Metal Hydride Batteries

The nickel metal hydride (Ni-MH) battery consists of three components, namely, an anode of hydrogen absorbing alloys (metal hydride; MH), a cathode of nickel hydroxide (Ni(OH)_2_) and an aqueous potassium hydroxide (KOH; typically 5M) electrolyte. However, an alkaline solid polymer electrolyte based on polyethylene oxide blended with KOH and water is also possible but with a lower ionic conductivity. The metal hydride can hold hydrogen up to 1–7%. The role of the alkaline component (which is KOH) is to boost the poor ion conductivity of the solid polymer. Nevertheless, the biggest challenge of the Ni-MH system is due to its inherent corrosion rate, which stems from the presence of oxygen. Generally, oxygen is evolved under overvoltage conditions and leads to a significant increase in cell resistance, which results in low conductivity and slower electrode kinetics due to hydroxide containing passivating layers [12]. Another issue is the requirement of high volumes of hydrogen gas [13]. Ni-MH is an environment friendly battery because it does not have toxic cadmium as seen in the Ni-Cd system. Having said that, Ni-MH lacks the capability of delivering high discharge power rates and is less tolerant to overcharge and fast charging compared to a nickel-cadmium battery. Ni-MH also has a very high self-discharge rate and relatively short cycle life compared to Ni-Cd [5,14].

Nickel hydroxide with a chemical formula Ni(OH)_2_, is a complex crystalline material, green in color, odorless, solid at room temperature, and nonhygroscopic in nature having a low solubility product. It is insoluble in water but soluble in acid and alkaline mediums. The nickel hydroxide exhibits electrical conductivity and it is a *p*-type semiconductor. It has two forms (a) beta-nickel hydroxide (β-Ni(OH)_2_), and (b) alpha-nickel hydroxide (α-Ni(OH)_2_). Distorted structures are deemed to be a pre-requisite for rechargeable battery applications. A. K. Shukla [14] discussed the various aspects of Ni-based rechargeable batteries including Ni-MH for battery applications. Furthermore, exploitation of nickel metal hydride batteries for electric vehicles and heavy electric vehicles is reported by Taniguchi [15]. Li Hui et.al. [16] then examined the electrochemical performances of nickel hydroxide such as charge-discharge storage capacities of this system. Based on all these studies, it is postulated that the following chemical reactions, Equations (1)–(3), occur at the positive electrode and negative electrodes along with their standard potentials (E^o^). During discharge, NiOOH is reduced to Ni(OH)_2_, and the metal hydride (MH) is oxidized to metal alloy (M) with the respective standard potentials shown in the Equations (1)–(3) vs. the standard hydrogen electrode. These electrochemical reactions in aqueous KOH electrolyte are reported to be reversible and therefore termed as secondary battery.
Positive electrode: Ni(OH)_2_ + OH^−^↔ NiOOH + H_2_O + e^−^, E^o^ = 0.490 V(1)
Negative electrode: M + H_2_O + e^−^↔ MH + OH^−^, E^o^ = −0.830 V(2)
Overall reaction: Ni(OH)_2_ + M ↔ NiOOH + MH, E^o^ = 1.32 V(3)

### 2.1. Performance Characteristics of Ni-MH Battery

The charge characteristics of commercial model Ni-MH batteries are given in Figure 1. The effect of varying the charge rate on Ni-MH battery reflects a smaller C-rate, and higher temperature possesses higher storage capacity. The obtained high-volume capacity is the most prominent benefit of using Ni-MH batteries [17,18] for wide applications. The charge-discharge profiles of the Ni-MH pouch cell with the effect of varying the charge rate is shown in Figure 2. In the typical galvanostatic plots, cathode material possessing a γ-phase inhibitor is reported to undergo a transformation from β-Ni(OH)_2_ to β-NiOOH and the process is reversible during the charge and discharge processes; while for the γ-phase promoter, the cathode material undergoes a distinct phase transformation from α-Ni(OH)_2_ to γ-NiOOH with two different charge voltage plateaus as depicted in Figure 2. Moreover, the pouch cells confirm that a lower charge rate results in higher charge and discharge capacities, and a lower operation temperature results in lower charge and discharge capacities. The two different configurations of the Ni-MH batteries shown in Figure 1 displayed same scenario regardless of their models. The specification of a commercial Ni-MH battery is given below:Energy/size: 140–300 Wh/LEnergy/weight: 40–80 Wh/kgPower/weight: 250–1000 W/kgSelf-discharge rate: 30%/monthCharge/discharge efficiency: 66%Nominal cell voltage: 1.32 VCycle durability: 500–1000 cycles

Apart from the battery configurations, according to the needs of consumer devices, different interior designs/specifications of Ni-MH batteries are also available [21,22]. They are classified based on their design of the positive (cathode) plate.

#### 2.1.1. Pocket Plate Technology

In this method, the active material is pelletized with a binder and conductive additive, and then to serve as current collector the pellets were placed in a perforated nickel-plated steel sheet (pocket specification). The flat pocket type [23] of construction consists of filling steel sheet-like pockets produced with perforated metal with nickel or cadmium oxide and arranging these pockets in the form of positive and negative plates. This sort of configuration is still used for low discharge rate applications but the sintered plate method (described in Section 2.1.3) offers enhanced electrochemical characteristics delivering at faster discharge rates with improved energy density.

#### 2.1.2. Tubular Plate Technology

The subsequent major improvement in the nickel electrode structure was tubular plate technology. In this system, perforated nickel-plated mild steel tubes were alternatively filled with layers of nickel flakes/graphite and nickel hydroxide. The primary objective of this technology is to ease the mechanical forces that result from the swelling of the active cathode material while extending the long-term stability of the battery during high depth-of-discharge cycling [5,14]. Tubular plates can have a variety of different types of porous tube materials and tube configurations and can have tubes joined together or formed as separate tubes which are located separately on the current collecting elements of the cell. The individual tubes were grouped in parallel to form the tubular plate electrode. As explained, due to the difficulties involved in the manufacturing process, this technology is no longer in use [5,14].

#### 2.1.3. Sintered Plate Technology

This sort of improved version is considered as an important milestone in the development of Ni-MH battery electrodes. In this system, the positive active material (Ni(OH)_2_) is deposited into the pores of a sintered plate or, in other words, porous metal supports. The plate is made by sintering carbonyl nickel powder into thin and highly porous plates at high temperatures, around 1000 °C, in a controlled atmosphere, leading to about 80% porosity, which is then impregnated with the active materials [24,25,26]. The nickel plate impregnated with nickel oxide (Ni(OH)_2_) is used as the positive plate, whereas the nickel plate impregnated with cadmium oxide is used as the negative plate. Sintered plates were fabricated using either a dry powder process or by a wet-slurry process. This kind of loose electrode can be modified to get the required balance of particle sizes, morphology, large surface area, and other mechanical properties such as adhesion and strength. Additionally, the sintered electrodes with improved material properties can be scaled up which is attractive for large scale applications. The charge process of Ni(OH)_2_ depends on the intrinsic properties of the electrode. The sintered plate type of battery has a much higher capacity and discharge rate compared to its earlier version of the Ni-MH battery technologies [5,14].

#### 2.1.4. Plastic Bonded Electrodes

Recently, efforts have been made to develop low cost cells for vehicular tractions. This resulted in the development of several plastic bonded nickel hydroxide electrodes. These electrodes are made by combining nickel hydroxide with a graphite conductive diluent and a plastic binder and pressing the mixture on to a screen current collector. This reduces the nickel requirements by about 40%. Excellent nickel hydroxide utilization has been obtained. These electrodes are very useful for structural determinations since interferences from the current collector are avoided. Later, other types of electrodes were developed for nickel-based batteries, they are (a) pasted nickel electrodes, (b) foam nickel electrodes, and (c) fiber nickel electrodes, et cetera. On the other hand, a polymer polypyrrole templated Ni(OH)_2_ in the form of nanowire configuration for energy storage has been reported [27] and it demonstrated a high storage. However, their charge propagation and mass production of cathode material is not found to be comparable to that of traditional Ni-MH batteries.

### 2.2. The Chemistry of Nickel Hydroxide

#### 2.2.1. The Structure of Nickel Hydroxide and Its Various Forms

The two polymorphs of nickel hydroxides with various type of disorders, namely, β-NiOOH and γ-NiOOH formed due to oxidation of β-Ni(OH)_2_ and α-Ni(OH)_2_, respectively. The chemical and electrochemical processes that occur in a Ni-MH battery are depicted in the general scheme, as shown in Figure 3 [28,29]. α-Ni(OH)_2_ could dehydrate in concentrated alkali medium into β-Ni(OH)_2_ and when the electrode has been overcharged, β-Ni(OH)_2_ could be transformed into γ-NiOOH.

Perfect crystalline β-Ni(OH)_2_ may not be suitable for battery applications or in other words, not a battery grade material. Water molecules, cations, anions, structural defects, et cetera, are introduced to β-Ni(OH)_2_ to modify/alter the crystallinity and to increase the electrical properties. Even though α-Ni(OH)_2_ has higher energy capacity, it is not preferred for use in commercial batteries, the main reasons being the long term stability problem, low gravimetric and volumetric energy densities, et cetera. Ramesh et al. [30] studied the effect of crystallite size in stacking faulted nickel hydroxide and its electrochemical performance. The X-ray diffraction (XRD) pattern of the products obtained from thermal decomposition of nickel hydroxide has also been reported in the literature. Ramesh, Kamath, and Shivakumara [31,32,33] studied the preparation of nickel hydroxide and the influence of precipitation conditions on structural disorder and their phase selectivity has also been detailed.

#### 2.2.2. β-Ni(OH)_2_

McBreen [9,10] reported that the structure of Ni(OH)_2_ consists of a hexagonal close packing of OH^−^ ions (as evidenced in Figure 4) with alternative layers of octahedral sites filled by nickel ions leading to stacking of charge-neutral layers. Bonding in Ni(OH)_2_ is anisotropic. The interlayer bonding is strongly iono-covalent in nature and in between the layers bonding is due to electrostatic interaction. Therefore, disorders have been formed due to the tendency of layers oriented with respect to each other which have been identified as an important source of improvement in electrochemical activity of nickel hydroxide.

#### 2.2.3. α-Ni(OH)_2_

α-Ni(OH)_2_ has an extremely hydrated turbostratic structure. Generally, these materials are semi crystalline in nature and sometimes exhibit no diffraction patterns. This hydrated nickel hydroxide (illustrated in Figure 5) has the common molecular formula of Ni(OH)_2_·(0.5–0.7H_2_O). The proposed structure for α-Ni(OH)_2_ is similar to the β-Ni(OH)_2_ structure, except that water molecules were present between the (001) planes, resulting in an expansion of the c-axis spacing to about 8 Å. A unit cell 3Ni(OH)_2_·2H_2_O, has also been proposed and assigned definite positions to the intercalated water molecules. The unit cell dimensions given by this model were a0 = 5.42 Å and c0 = 8.51 Å. The comparative structure for α-and β-Ni(OH)_2_ is shown in Figure 5. The water molecules separate the layers of α–nickel hydroxide which are H-bonded to the Ni-OH groups.

#### 2.2.4. β-NiOOH

It is comparatively amorphous in nature and bears a distorted brucite structure exhibiting a hexagonal lattice. The dimensions of unit cell change of β-Ni(OH)_2_ from a0 = 3.13 Å to 2.82 Å and from c0 = 4.60 Å to 4.85 Å. The infra-red spectral analysis of this material indicates that NiOOH is a hydrogen-bonded structure with no free OH^−^ groups.

#### 2.2.5. γ–NiOOH

This phase corresponds to NiO_2_ layered structures. The spacing between the Ni layers is 7.2 Å but it depends on the insertion of alkali cations and water molecules. Therefore, transition of β-Ni(OH)_2_ to γ-NiOOH, results in swelling due to large volume expansion, whereas α/γ couple provides an advantage.

### 2.3. Properties of Nickel Hydroxide

#### 2.3.1. Discharge Capacity

Nickel has two oxidation numbers, namely, +2 and +3. In the reduced form of nickel hydroxide, either in β- or α-phase the nickel exhibits a +2 oxidation state. When β-Ni(OH)_2_ converts to β-NiOOH by the reaction (4), protons are extracted (proton diffusion) from the solid-state brucite lattice. The oxidation state of nickel at β-NiOOH (oxidized form) becomes +3, by one electron transfer.
β-Ni(OH)_2_ + OH^−^ ⇔ β-NiOOH + H_2_O + e^−^, E^o^ = 0.49 V (SHE)(4)

Accordingly, the theoretical discharge capacity becomes 289 mAh/g of nickel hydroxide. When β-Ni(OH)_2_ on extended charging or α-Ni(OH)_2_ on charging forms γ-NiOOH, the nickel’s oxidation state is +3.7 (an empirical formula of H_0.3_NiO_2_), the reaction remains the same as (4) with transfer of 1.3 e^−^ per atom of Ni(OH)_2_. As shown in the general scheme (Figure 3), γ-NiOOH has been electrochemically reversible with α-Ni(OH)_2_, but not with β-Ni(OH)_2_. During α ⇔ γ phase conversion, the number of electrons replaced per nickel atom is high and hence nickel positive electrodes consisting of α-Ni(OH)_2_ are expected to have a larger theoretical capacity than β-Ni(OH)_2_. Here, the theoretical discharge capacity becomes 456 mAh/g of nickel hydroxide.

#### 2.3.2. Density

Another important characteristic of Ni(OH)_2_ is the bulk density, which determines the loading capacity within a battery. The theoretical bulk density of α-Ni(OH)_2_ is 2.82 g/cc, whereas it is 3.97 g/cc for β-Ni(OH)_2_. Thus β-Ni(OH)_2_ is being utilized at present with a discharge capacity of very near to its theoretical capacity having a tap density of 2.1 g/cc. If the theoretical capacity of the α-Ni(OH)_2_ could be amply utilized and its stability in strong alkali can be ensured, it would counterbalance for the less tap density.

#### 2.3.3. Ni-Based Electrocatalyst for Oxygen Evolution Reaction

Energy storage will be a critically important part of the transition to a low-emissions future. Currently there is a heavy reliance on fossil-fuelled energy for both baseload and as a backup for times when renewable output decreases suddenly. Hence, developing clean and sustainable energies and decarbonization are vital for our planet. Clean and green energy such as hydrogen generation using renewable energy resources such as wind and solar as inputs to split water electrochemically into hydrogen and oxygen is potentially attractive on a commercial scale. However, the associated oxygen evolution reaction (OER) from the anode electrode is the efficiency limiting process. Platinum group materials are widely known to be used as OER catalysts, but it is expensive. Therefore, developing cost-effective, highly active, and stable catalysts for OER is crucial. Recently, Sun et al. [34] reported the progress on nickel-based oxide (NiO) and oxyhydroxide (Ni(OH)_2_) composites for water oxidation catalysis through materials design/synthesis and electrochemical performance. The modifications made to these materials are reported in this review article to enhance the oxygen evolution reaction (OER) mechanism and their nanoarchitectures exhibited highly efficient catalysts. Future research trends and perspectives on the development of nickel-based oxygen evolution reaction electrocatalysts are discussed further by Boetcher’s group [35]. They have studied the incorporation of Fe in the improvement of the catalytic activity of nickel hydroxide, which is correlated to the local structure. Apart from nickel-based catalysts, nickel-based compounds have also been reported as the most promising earth abundant OER catalysts attracting ever increasing interest due to high activity and stability [36]. In this work, molybdenum (Mo) and iron (Fe) modification appear to have a synergistic effect to enhance both the OER activity and stability with no degradation after 50 h and are shown to have outperformed all OER catalysts reported. Overall, nickel oxide as a non-noble metal OER has shown potential in favor of the OER process.

## 3. Production of Nickel Hydroxide

The major producers of nickel hydroxide are Ovonic Battery Company (Troy, MI, USA), Inco Company (Toronto, ON, Canada), NEXcell Battery Co., Ltd. (Hsinchu City, Taiwan), et cetera. In commercial practice, nickel hydroxide is prepared by chemical precipitation method. The majority of companies follow a low-pressure and room temperature precipitation process with controlled parameters such as salt and caustic concentration, pH, temperature, agitation speed, et cetera. The Ovonic battery company is producing battery grade nickel hydroxide from nickel sulphate, ammonium hydroxide, and sodium hydroxide solutions that continuously enter the reactor. The nickel hydroxide and effluent have been collected from an overflow at the side of the reactor. The sample obtained after filtration and water washing was dried at 175 °F for 2 h, which is then ready to be used.

### Criteria for Preparation

Two properties such as the reversible discharge capacity and the tap density are highly significant for battery grade nickel hydroxide, because they determine the power capacity and capacity density of the battery. A good crystalline β-Ni(OH)_2_ is not electrochemically active [37]. In reality, the reversible discharge capacity of Ni(OH)_2_ is dependent on phase, structural characteristics such as the crystalline lattice disorders, degree of crystallinity, crystallite size, orientation, crystal growth, anions, and H_2_O adsorbed, and intercalation in crystals. These properties are strongly dependent on the process of synthesis. Therefore, the primary focus of the researchers has been to control the synthesized parameters to commercialize the product. Synthesis of Ni(OH)_2_ is a straight-forward phenomenon described by Equation (5) given below.
Ni^++^ + 2OH^−^ → Ni(OH)_2_↓(5)

This reaction can take place in various ways. It can be summarized into two categories, namely, chemical and electrochemical methods of preparation. Further, Figure 6 shows the stability domain of nickel hydroxide with respect to Eh and pH values. Table 2 summarize the formation of Ni structure at various pH values. Ravi Kumar et al. [38] studied the precipitation of nickel hydroxide at different pH values and their X-ray diffraction structural studies showed the samples prepared at higher pH values of around 10 indicating an increased degree of ordering and crystallinity forming β-Ni(OH)_2_ material. To precipitate Ni(OH)_2_, the desired pH and bath potential need to be maintained at the production site.

## 4. Technologies for Preparation of Battery Grade Nickel Hydroxide

The development of technologies for preparing high quality nanocrystalline materials has offered an opportunity to improve the electrochemical properties of nickel hydroxide and commercialize it. A few of the key reports are quoted in this section. Dima et al. [40] studied the reduction of nitrate electrocatalytically at low concentrations on coinage and transition-metal electrodes in acid solutions. Dong et al. [41] fabricated porous nickel oxide and nickel hydroxide nanostructures having various morphologies under controlled conditions. Xu et al. [42] studied Ni (II) complexation to amorphous hydrous ferric oxide with an emphasis on using the X-ray absorption spectroscopy technique. The performance of multiphase nano-structural nickel hydroxide was studied by Wang et al. [43]. Characterization studies of β-nickel hydroxide obtained from supersonic co-precipitation method was reported by Xu et al. [44]. Structural features of nickel hydroxide obtained from a chemical precipitation method under various pH values was done by Song [45]. All of the studies highlighted the nano crystallinity of nickel hydroxide and their useful properties.

### 4.1. Chemical Method of Preparation

Nickel hydroxide is a low solubility product and hence, can be precipitated by providing an alkali (OH^−^) and a suitable nickel salt (Ni^++^) under proper conditions [28,30]. This precipitation can be achieved by (a) strong alkali, (b) ammonia, (c) urea hydrolysis, and (d) hydrolysis of nickel acetate using a hydrothermal route. The parameters that affect the product quality are (i) pH (ii) reaction temperature, (iii) choice of alkali, (iv) nickel salt, (v) mixing rate, (vi) ageing, (vii) subsequent hydrothermal treatment, et cetera.

#### 4.1.1. Alkali Induced Precipitation

Generally, a strong alkaline medium such as sodium hydroxide (NaOH) or potassium hydroxide (KOH) is chosen to precipitate nickel hydroxide from a nickel salt solution.
NiSO_4_ + 2NaOH → Ni(OH)_2_↓ + Na_2_SO_4_(6)

In many cases ammonium hydroxide (NH_4_OH) and lithium hydroxide (LiOH) are also used for the purpose. Among the available different nickel salts, nickel sulphate, nickel nitrate, and nickel chloride have been extensively used as source material for nickel. Different attempts such as (a) nickel salt addition to alkali solution, (b) addition of alkali to a solution of nickel salt, or (c) alkaline and nickel salt solutions fed into the reaction tank at the same time have been made to prepare nickel hydroxide. A much better quality product could be achieved with a tap density of about 1.77 g/cc and a specific capacity of 0.252 mAh/g, when nickel sulphate and sodium hydroxide are fed into the reaction vessel simultaneously at a temperature of 110 °C, pH of about 12.0–12.2, and the product is dried at a temperature of 110 °C. Drop wise addition of potassium hydroxide to nickel sulphate or nickel nitrate at a flow rate of 10 mL/min under constant stirring condition produces a mixture of α and β-nickel hydroxide, whereas, uncontrolled addition of potassium hydroxide to nickel nitrate generates only α-nickel hydroxide. The nickel hydroxide precipitated from nickel nitrate using lithium hydroxide in the presence of aluminum is of α-phase with a high discharge capacity of 380–400 mAh/g. The β-nickel hydroxide product with zero moisture content could be produced from nickel sulphate and nickel chloride by a special spraying technique, using ammonium hydroxide and potassium hydroxide, which gave a discharge capacity of 278 mAh/g and showed smaller crystallite size. The effect of pH, temperature, alkali concentration, and various nickel sources have been investigated by Ramesh and Kamath [30]. Nickel nitrate addition to a sodium hydroxide pool has been carried out at a controlled rate of 4 mL/min. Temperature and pH have effect on the phase, structure, and structural disorder of the nickel hydroxide. The crystallinity of the product has also been found to be affected adversely when a non-nitrate source of nickel is used. Another report [41] has shown the production of nano sized α-Ni(OH)_2_ of various morphologies by a hydrothermal method under mild conditions using NaOH, NiSO_4_, and water.

#### 4.1.2. Ammonia Induced Precipitation

Ammonia has been extensively used as a precipitating or complexing agent in the synthesis of nickel hydroxide. Invariably, β-nickel hydroxide with different electrochemical properties formed with moderate to no water content. A discharge capacity of 248 mAh/g is found when nickel hydroxide is precipitated from nickel sulphate, sodium hydroxide, and ammonium hydroxide at controlled pH and temperature conditions under vigorous stirring. Attempts have also been made to prepare α-nickel hydroxide by using nickel nitrate and sodium hydroxide as raw materials in the presence of a buffer solution made from ammonium hydroxide and ammonium chloride. The sample shows a discharge capacity of 303 mAh/g. However, homogeneous precipitation methods applied to precipitate β-nickel hydroxide from nickel nitrate and ammonia solution results in a discharge capacity of 260 mAh/g while β-nickel hydroxide precipitated by nickel sulphate, sodium hydroxide, and ammonia gives a discharge capacity of 275 mAh/g at a discharge rate of C/5. In another case [43], nickel hydroxide is prepared in a two-step process, where, nickel sulphate is mixed with ammonium hydroxide in a specific amine reactor to form nickel ammonium complex, which is then removed and combined with sodium hydroxide in a second reactor. In another report [44], nickel hydroxide is prepared of size 8–12 µm and tap density 1.8–2.3 g/cc by electroless precipitation in a single reactor using a continuous and simultaneous supply of reactants with rapid stirring.

#### 4.1.3. Homogeneous Precipitation by Urea Hydrolysis

Urea has been utilized as a source of ammonia for precipitation of nickel hydroxide [46]. It provides ammonia in a constant and uniform rate in the mother liquor for homogeneous precipitation. The urea hydrolysis precipitation reaction may be given as:NH_2_-CO-NH_2_ + NiSO_4_ + 3H_2_O → Ni(OH)_2_↓ + (NH4)_2_·SO_4_ + CO_2_(7)

Pure α-Ni(OH)_2_ has also been synthesized by high concentration of urea at a lower temperature of 90 °C or by low concentration of urea at a higher temperature of 120 °C.

#### 4.1.4. Other Precipitation Methods

A mixed phase of α and β-nickel hydroxide has been precipitated from nickel sulphamate and nickel ammonium sulphate with sodium hydroxide. α-nickel hydroxide with a considerable water content could be produced from nickel acetate in glycine and n-butanol with KOH in butanol or in dodecyl sulphate with ammonia. Sodium oxalate as complexing agent in ethanol solvent could produce β-type nano nickel hydroxide with average size range of 5–35 nm. The average discharge capacity has been within 140 mAh/g. Nano phase β-nickel hydroxide with a discharge capacity of 289 mAh/g could be prepared from nickel sulphate, sodium hydroxide with some dispersant, and chelating reagent. Nano phase β-nickel hydroxide has also been prepared by using raw materials such as nickel nitrate and sodium hydroxide in the presence of ethelenediamine. Nickel hydroxide prepared in the presence of complexing agents such as tri-ethanol amine, ethylene-diamine tetra acetic acid, sodium potassium tartrate, sodium succinate, et cetera, has a discharge capacity of 208 mAh/g. Trivalent anions like carbonate has been introduced [37] from sodium carbonate to the nickel hydroxide lattice for stabilization of the highly hydrated α-phase with improvement in discharge capacity during the chemical precipitation method.

#### 4.1.5. Subsequent Treatment of Precipitated Nickel Hydroxide

Various operations such as continuous agitation, hydrothermal treatment, ultrasonic dispersion, ageing in mother liquor, ball milling or pulverization, et cetera, have been applied either during or subsequent to chemical preparation of nickel hydroxide. Much attention has been diverted to hydrothermal treatment of precipitated nickel hydroxide for enhancement in electrochemical properties. The α-nickel hydroxide precipitated from nickel nitrate with ammonium hydroxide has been treated hydrothermally to get highly efficient β-nickel hydroxide. Hydrothermal treatment to nickel nitrate and lithium hydroxide in the presence of aluminum could produce extremely fine grained good crystalline nickel hydroxide having a discharge capacity of 400 mAh/g. Nano phase spherical β-nickel hydroxides have been obtained by low temperature hydrothermal treatment to nickel sulphate, ammonia, and sodium hydroxide solution, as well as by dropping 0.05 m/L KOH solution to a nickel nitrate pool (0.025 m/L) under stirring with simultaneous ultrasonic dispersion condition. The discharge capacity of the resultant samples is 260 and 381 mAh/g, respectively. Pure α-nickel hydroxide with a tap density of 0.75 g/cc and BET surface area 20.1 m^2^/g has been produced in an autoclave. Under hydrothermal conditions, using ethanol, the β-nickel hydroxides with single crystal structure having 2.0 ± 0.5 nm diameter and 22 ± 5 nm length were obtained. The nickel hydroxide structure has been modified with high energy ball milling. In another study it has also been observed that normal ball milling increases the surface area, decreases the particle and crystallite size, and reduces the crystallinity of β-nickel hydroxide which is advantageous in increasing the electrochemical performance of nickel hydroxide. Properly ordered mesoporous hydroxide thin films with self-assembly of size-tailored nano building blocks was reported by Naoki Tarutani [47]. Arshid Numan [48] synthesized nickel hydroxide of nano-size and X. P. Wang [49] studied strain stabilized nickel hydroxide nano ribbons for effective water splitting.

### 4.2. Electrochemical Method of Preparation

Nickel hydroxide can also be prepared through electrochemical route [50,51,52,53,54,55,56,57,58,59,60,61,62]. This way of preparation provides some advantages such as purity of the product, elimination of some anions and cations in the system, better control of the process through regulation of current density, and environment friendliness of the process. It has already been established that nitrate ions reduce to nitrite ions or ammonium ions in the cathode chamber to produce hydroxyl ions during electrochemical synthesis of nickel hydroxide by using nickel nitrate as raw material. Production of hydroxyl ions increase the local pH and precipitate nickel hydroxide reacting with nickel ions. Simultaneously, oxygen evolution reactions occur at the inert iridium coated titanium electrode at the anode chamber resulting in a decrease in local pH by production of H^+^ ions. A diaphragm cell is used to stop the direct intermixing of the highly acidic anolyte with highly basic catholyte. Within a space of time various attempts have been taken to precipitate battery grade nickel hydroxide. Some of them are presented below.

#### 4.2.1. Reactions for Electrochemical Precipitation of Nickel Hydroxide

As mentioned in the previous section, reaction (5) remains as such for the cathode. The anodic reactions are given in Equation (8) for nickel anode and Equation (9) for inert anode.
Ni → Ni^++^ + 2e^−^, E^o^ = −0.257 V (SHE)(8)
2H_2_O → O_2_↑ + 4H^+^ + 4e^−^, E^o^ = 1.23 V (SHE)(9)

Nickel nitrate is employed as the electrolyte and the nitrate ion is reduced to produce hydroxyl ions at the cathode which increase the pH close to the cathode to precipitate Ni(OH)_2_ as per reaction (5) using nickel ions present in the bath. However, there are several reports on the exact reaction. Nitrate reduction to nitrite as per the following reaction has been postulated.
(10)$$NO_{3}{}^{-}{+H}_{2}O+2e^{-}\to NO_{2}{}^{-}+2OH^{-},{\;E}^{o}=0.01\;V\;(\mathrm{SHE})$$

Another reaction has been proposed where the following reaction at a current density of 5 mA/cm^2^ was quantitatively conformed.
(11)$$NO_{3}{}^{-}{+10H}^{+}+8e^{-}\to NH_{4}{}^{+}+3H_{2}O$$

Subsequently, other reports showed the reaction at higher current densities (10, 30 and 60 mA/cm^2^) and suggested the following reaction.
(12)$$NO_{3}{}^{-}+7H_{2}O+8e^{-}\to NH_{4}\mathrm{OH}+9OH^{-},{\;E}^{o}=-0.12\;V\;(\mathrm{SHE})$$

The same reaction with splitting of NH_4_OH into ions as per reaction (12) has been reported widely. Instead of $NH_{4}{}^{+}$, NH_3_ has also been postulated as the reaction (11) product. Later, the ammonia/ammonium-ion equilibrium, as expressed by reaction (12) which moves towards the right hand-side in acidic and neutral conditions, converts reaction (11) to reaction (10). Other proposed reactions are:(13)$$NO^{-}+7H_{2}O+8e^{-}\to NH_{4}{}^{+}+10OH^{-},{\;E}^{o}=-0.12\;V\;(\mathrm{SHE})$$
(14)$$NO_{3}{}^{-}{+6H}_{2}O+8e^{-}\to NH_{3}+9OH^{-},{\;E}^{o}=-0.12\;V\;(\mathrm{SHE})$$
(15)NH3 + H2O ↔ NH4+ + OH−
(16)$$NO_{2}{}^{-}+5H_{2}O+6e^{-}\to NH_{3}+7OH^{-},{\;E}^{o}=-0.165\;V\;(\mathrm{SHE})$$
(17)$$2NO_{2}{}^{-}+4H_{2}O+6e^{-}\to N_{2}+8OH^{-},{\;E}^{o}=0.406\;V\;(\mathrm{SHE})$$
(18)$$2NO_{2}{}^{-}{+3H}_{2}O+4e^{-}\to N_{2}O+6OH^{-},{\;E}^{o}=0.15\;V\;(\mathrm{SHE})$$
(19)$$NO_{2}{}^{-}{+6H}_{2}O+6e^{-}\to NH_{4}{}^{+}+8OH^{-}$$

However it has been observed that reactions (14) and (15) combine to result in the Equation (13), reactions (15) and (16) may combine in a similar fashion to give reaction (19).

Regarding reaction (5), at higher concentration of (Ni^++^) in the bath, there is an observation that it takes place in two steps as follows:(20)$$4Ni^{++}+4OH^{-}\to Ni_{4}{(OH)}_{4}{}^{+4}$$
(21)$$Ni_{4}{(OH)}_{4}{}^{+4}+4OH^{-}\to 4Ni{(OH)}_{2}$$

The stability diagram presented in Figure 7 supports this hypothesis.

The previous discussions indicated that the reduction of nitrate ions at the cathode is the central process in the electrolytic preparation of nickel hydroxide. It has been shown by the authors that [50,51,52,53,54,55,56,57,58,59,60,61,62] through cyclic voltammetry that the main reaction in the process is electrochemical reduction of nitrate to nitrite followed by an irreversible chemical reaction to form nickel hydroxide. In some studies, it has been observed that the reaction mechanism depends strongly on the electrode potential, electrolyte pH, nature of electrode material, and presence of additives in the solution. In a bulk electrolysis, millimoles of nitrate mediated by under-potential-deposited cadmium on silver at pH = 3 and E^o^ = −0.5 V (SCE), the predominant product is nitrite. At pH = 1, the reduction goes beyond nitrite to NO, N_2_O, N_2_, NH_2_OH, NH_3_, et cetera. Cattarin [55] has demonstrated that nitrate reduction in 1 M NaOH supporting electrolyte gives nitrite at −1.4 V (SCE) on the Ag cathode. Nitrite is reduced to ammonia at potential 500 mV more negative. These two reduction steps take place at −1.1 V (SCE) and −1.4 V (SCE) on the Cu cathode.

#### 4.2.2. Electrolytic Preparation of Nickel Hydroxide Thin Films

Many fundamental studies are available in literature [47,56] on electrochemically prepared thin films electrodes of nickel hydroxide on platinum or other metals. Most of these studies deal with influence of additives on the properties of nickel hydroxide. Thin films are easy to develop in a few minutes on a gold or nickel substrate and immediately the electrochemical studies can be carried out. Nickel and cobalt hydroxides were deposited in an approximate ratio of 9:1 onto a nickel foil from a solution containing 0.01 M Co(NO_3_)_2_ + 0.09 M Ni(NO_3_)_2_ at a current density of 16 mA/cm^2^ in a few seconds. Subsequently, they characterized the film by cyclic voltammetry.

#### 4.2.3. Nickel Hydroxide Electrodes by Electrochemical Impregnation

A wide range of investigations have been conducted on electrochemical impregnation of porous nickel sintered plates with nickel [59]. The reason was to use sintered nickel plaque loaded with nickel hydroxide acting as the positive electrode in the battery. The electrochemical impregnation method consists of the cathodic reduction of nitrate ions within the pores of the porous nickel plaque cathode to generate hydroxyl ions which react with nickel ions to precipitate Ni(OH)_2_ in the pores. The loading level and the plaque expansion assume great significance. Uniform coating of the active material is very important which can be achieved by (i) increasing the bath temperature, (ii) decreasing the applied current density, (iii) maintaining appropriate electrolyte composition (1.5–3.0 m/L nickel nitrate and a small amount of sodium nitrite), and (iv) gradually reducing the current density during the process. A “Fibrex Mesh” and 4 M nickel nitrate concentration were observed to be the most suitable. In another study, significant decrease was observed in the rate of deposition with an increase in (Ni^++^). It has been linked to formation of $Ni_{4}{(OH)}_{4}{}^{+4}$. The effect of temperature and the ethanol content was also established.

In general, the electrochemical impregnation is achieved through the following approaches:(i)Low current density (5 mA/cm^2^) with low (Ni^++^) (~0.3 M) and a soluble anode,(ii)High current density (175 mA/cm^2^) with (Ni^++^): 4 M,(iii)Nickel nitrate and cobalt nitrate dissolved in 1:1 ethanol-water mixture and(iv)Ni(NO_3_)_2_ + Co(NO_3_)_2_ + NaNO_2_ at pH = 3 with platinized platinum, anode.

Lastly, the nickel sintered electrodes have been replaced with nickel foam electrodes, because, the latter achieve a capacity density up to 700 mAh/cc in comparison to 450 mAh/cc by the former.

#### 4.2.4. Bulk Production of Nickel Hydroxide by Electrochemical Method

Unlike the chemical method, the electrochemical procedure for synthesis of nickel hydroxide has not been adopted commercially. A patented work describes the principle, that the anodic oxidation of metallic nickel in aqueous electrolyte containing both sulphate and chloride leads to a nickel hydroxide intermediate product which can be converted into pure hydroxide by treatment with alkaline hydroxides. Accordingly, in a two-step process, nickel hydroxide could produce with 1% Co and less than 500 ppm anions with tap density of 1.8 g/cc, BET surface area of 88 m^2^/g, and full width at half-maximum (FWHM) of (001) plane limited to a value of 0.6° in two theta degrees. The presence of Co appears to affect the valence state of Ni^2+^ reversibility in the electrochemical process. In another method, nickel hydroxide produced by electro-dialysis in a membrane bipolar cell. The pH in these two cases was more than 7. Nickel hydroxide was precipitated from a nickel nitrate solution maintained at pH 1.8–2.2, by 1 M sodium nitrate, at a current density of 50–200 A/m^2^ and a temperature of 80–95 °C. The authors have reported [50,51,52,53,54] electrolytic preparation of nickel hydroxide. They optimized the process parameters and got β-Ni(OH)_2_ with a discharge capacity of 200 mAh/g.

As discussed in the introduction section, discharge capacity and power density are the main parameters of the nickel hydroxide positive electrode materials. α-Ni(OH)_2_ and its stability are of great significance in determining the amount of the charge that can be stored reversibly during the electrochemical processes. Besides, increasing the conductivity of the nickel hydroxide and managing the parasitic oxygen evolution reaction during the oxidation stage are also important to operate for multiple cycles with capacity retention. Additives, or in other words, modifiers, also have a prominent role to play. Thus, the main function of additives in nickel hydroxide electrodes are (a) to enhance the utilization factor and activity of active materials, (b) to raise the electrode conductivity, (c) to inhibit the γ-phase formation and raise the operating time, (d) to increase the oxygen separation polarization, (e) to rise the charging efficiency, (f) to decrease the quantity of oxygen gas separated out, et cetera. Some of the main additives used in the Ni-MH battery are Co, Cd, Mn, Zn, Al, Fe, Ca, Mg, Y, et cetera. The additives can be added to nickel hydroxide by co-substitution during preparation or post substitution. Co-substitution is found to be more beneficial as the substituted component mixes homogeneously with nickel hydroxide.

The authors have tried to find out the exact reaction mechanism of electrochemical synthesis of battery grade nickel hydroxide both by potentiostatic measurement and electro-crystallization method. To understand the process, refer to the cathode and anodic reactions stated in Equations (5), (9), (10) and (13) referred to in earlier sections.

A typical polarization plot at 2.04 M (${\mathrm{NO}}_{3}{}^{-}$), 1.02 M (Ni^++^) having pH = 3 and bath temperature 25 °C is shown in Figure 8. It is found from the plot that the anodic curve initiates from a rest potential of 0.501 V and passes nearly linearly (with an upward potential) up to a current of 0.1 A. Then the polarization plot exhibits a passivation process until 1.2 V/0.26 A. Subsequently, the transpassive region starts and continues up to 1.6 V/1.2 A with a Tafel slope of 0.550 V/decade. This portion may represent reaction (9), since it lies close to the potential represented by this reaction. Then the plot continues vertically at a limiting current of 1.3 A. On the other side, the cathodic region of the Tafel plot (in Figure 8) shows three regions (called I, II, and III) with the corresponding reactions.

Besides reactions shown in Equations (10) and (13), the other reduction reactions stated in the Equations (14)–(19) are also known to take place. For all these reactions the E^o^ values are close to the E^o^ values of reactions (10) and (13) and in fact, reaction (13) is the combination of reactions (10) and (19). Furthermore, NH_3_ generated in reaction (14) may combine with one molecule of H_2_O to produce ${\mathrm{NH}}_{4}{}^{+}+OH^{-}$, which is equivalent to reaction (13). It may also be possible that the nitrite ion produced by Equation (10), is consumed by Equation (19) to give products as in Equation (13). Thus, combination of some of these reactions may be possible.

#### 4.2.5. Quantitative Assessment of Electrochemical Precipitation of Nickel Hydroxide

The electrochemical precipitation of battery grade Ni(OH)_2_ synthesized galvanostatically at a current of 0.65 A and the occurrence of reaction during this process has been discussed in this section. The electrochemical conditions used for the synthesis are a stainless-steel plate cathode of 32.5 cm^2^ area and iridium coated titanium sheet anode of 37.5 cm^2^ area, 0.1 L each of 1.02 M nickel solution as anolyte and catholyte, were maintained. The anolyte and catholyte were kept at pH of 3. The experiment was conducted for a period of six hours.

The average cell voltage was found to be 3.41 V during electrolysis. The anode potential raises from 1.245 to 1.379 V and the cathode potential reduces from −0.929 to −1.223 V in six hours, along the vertical line denoted as AB, shown in Figure 8 with arrow marks. The pH of the anolyte reduces from 3 to 0.21 in half an hour and attains 0.1 at the end of six hours of electrolysis. In the first half an hour, the pH of catholyte rises from 3 to 6 and attains 6.4 after six hours. It may be observed that precipitation occurred instantly although the pH took some time to reach 6. The solubility/ionic product of nickel hydroxide may be determined as:K_SP_ = [Ni^++^]·[OH^−^]^2^(22)

When the value of ionic product of nickel hydroxide exceeds the solubility product, precipitation occurs. The values of ionic /solubility product at various times such as 0.5 h (K_SP_ = 0.5), 3 h (K_SP_ = 3.0) and 6 h (K_SP_ = 6.0) are calculated according to Equation (22). Various values of solubility product (K_SP_) for nickel hydroxide precipitation in aqueous solutions have been reported ranging from 1.6 × 10^−14^ to 6.3 × 10^−18^.

As far as nitrate reduction is concerned, the current efficiency is 72.23% with anhydrous Ni(OH)_2_ as product. The nickel hydroxide sample obtained from electrolysis was reported [50] to be characterized by SEM and XRD to know the morphological features and the phase of the product. The electron micrograph shows the formation of semi-spherical distinct particles of varied sizes with average diameter around 10 µm. It is observed from the XRD pattern that the sample obtained is an α- and β-type nickel hydroxide mixture. The peaks are not sharp, which indicates a poor crystalline material. Particle size analysis indicates d50 to be 21 µm, surface area is 133 m^2^/g and the tap density is 1.31 g/cc.

The properties of nickel hydroxide OH^−^/Ni^++^ with a ratio of above six and lower values are mentioned in Table 3 and Table 4.

It may be concluded that the rate of supply of OH^−^ plays a prominent role in determining the character of the Ni(OH)_2_ species.

Another scope of this study is that of change of anode material. During electrochemical synthesis of nickel hydroxide from nickel nitrate, nitrate ions reduce at the cathode to liberate hydroxyl ions and oxygen evolution occurs at the non-consumable anode. The E^o^ value for the later reaction is 1.229 V (SHE). If the above-mentioned Ti anode is replaced with nickel it would dissolve as per reaction (8). As a result, the anode potential and cell voltage decrease, which consequently decreases the energy consumption. Investigations of this issue show that current efficiency improved significantly, increasing the rate of supply of OH^−^ and hence the stated ratio. It has been observed that the ratio is improved in comparison to those with Ti non-consumable anode. Accordingly, β-Ni(OH)_2_ content and sharpness of the peaks representing better crystallinity with the nickel anode. The (OH^−^)/(Ni^++^), not only resulted in improving the quality but also increased the amount of β-Ni(OH)_2_. It has been observed that the rise in the ionic ratio increases the size and decreases the discharge capacity. However, the size of the precipitate significantly decreases with an increase in ionic ratio and the discharge capacity increases, which needs further explanation.

The reasoning may be considered related to the current efficiency. A mass balance with respect to coulombic efficiency for the experiment with Ti anode showed that the Ni(OH)_2_ precipitation could account for a current utilization of only 57.87%. As per an estimation presented, it could be found that almost equivalent amounts of OH^−^ in the cathode chamber and H^+^ in the anode chamber remained unaccounted for. Looking at the final pH of the catholyte and anolyte it was presumed that these two species might have combined, neutralizing their charges. In the experiments with nickel sheet replacing Ti anode, H^+^ production at the anode was replaced with Ni^++^ production and, hence, full consumption of OH^−^ in the cathode chamber took place. The current efficiency in both the chambers touched close to the 100% mark, as no H^+^ was produced in the anode chamber. OH^−^ from the cathode chamber whose diffusivity co-efficient is 5.6 × 10^−5^ cm^2^/s might have moved to the anode chamber and raised the pH to more than 5 from the value of 3, was observed.

## 5. Emerging Technology Developments and Future Scope

Structural disorders, which play a significant role in the electrochemical property of the nickel hydroxide, are yet to be studied. These are known to be affected through agitation, ageing, hydrothermal treatment, sonication, et cetera. Each of these aspects could form the topics of future research. Disorders in the crystal lattice of chemically precipitated β-Ni(OH)_2_ have been investigated through simulation studies. Similar studies are to be conducted with the sample produced through the electrochemical route. Further, such studies should be extended to α-Ni(OH)_2_ and the mixtures of α- and β-Ni(OH)_2_, which are going to form the major topics of future investigations. Energy density is another significant parameter which depends on the morphology, specifically the tap density. A lot of scope remains within the electrochemical preparation method to regulate the tap density of the materials. Therefore, with the improvement of Ni(OH)_2_ from a materials aspect, adding to the intrinsic safety of Ni-MH batteries using KOH electrolyte, they have plenty of opportunities in hybrid EVs. Having said that, Ni-MH batteries cannot compete with the current generation of lithium-ion batteries [63], and next generation sodium [64] and potassium-ion batteries [65] (shown in Figure 9), as they exceed both in terms of gravimetric and volumetric energy density.

## 6. Conclusions

Rechargeable alkaline nickel-metal hydride batteries (Ni-MH) have gained wide attention due to their low cost, safety, and excellent stability. However, with the optimized electrochemical route of producing (Ni(OH)_2_) cathode material, it has a great future in the battery industry. This perspective discussed the synthesis routes of producing nickel hydroxide chemically and electrochemically. Different precipitations were developed to produce nickel hydroxide chemically. A mixed phase α-Ni(OH)_2_ and β-Ni(OH)_2_ is being produced in the chemical method. The electrode reactions taking place during electrolytic production of Ni(OH)_2_ from nickel nitrate have been established. Among the three process variables such as the applied current density (rate of OH^−^ supply), initial (Ni^++^), and the initial pH, the influence of the first one on the characteristics of Ni(OH)_2_ is the most eminent followed by initial (Ni^++^). Effect of the initial pH is not significant; current density has been successfully employed as rate of OH^−^ generation in the bath through which OH^−^ to Ni^++^ ratio. A ratio of greater than 6 with reasonable (Ni^++^) produces good crystalline β-Ni(OH)_2_. A value of less than 1, produces α-Ni(OH)_2_. It has been demonstrated that the electrolytic preparation method holds great promise through its precise control of chemical and electrochemical parameters. It is up to the end users to choose the type of nickel hydroxide they opt for based on that the electrochemistry can be tweaked. The proposed electrolytic production method used is scalable and costs less, and the process is quicker than other counterparts of controlled morphology and other parameters.

## Figures and Tables

**Figure 1 nanomaterials-10-01878-f001:**
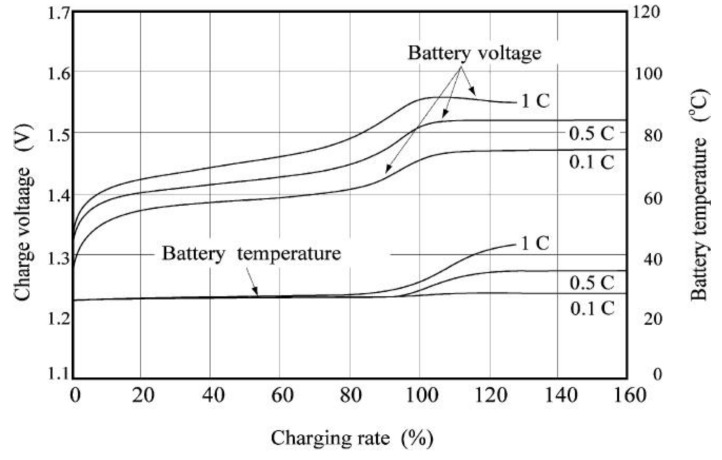
Charging characteristics of commercial Ni-MH batteries for a HR-4/5AU model. (charge: 1 °C for 1.3 h; 0.5 °C for 3.2 h; 0.1 °C for 16 h). Reproduced with permission from [19]; Elsevier, 2020.

**Figure 2 nanomaterials-10-01878-f002:**
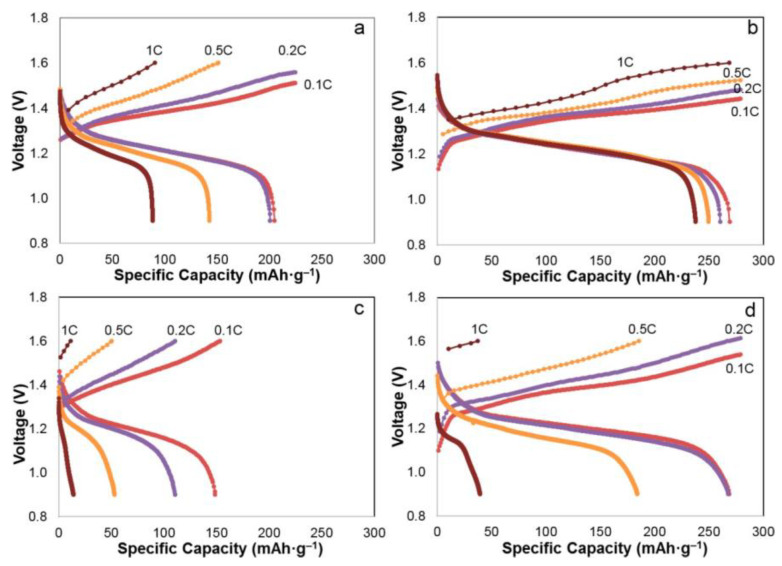
Charge-discharge characteristics of Ni-MH batteries at various current rates. (**a**,**c**) γ-phase inhibitor additive in Ni(OH)_2_ cathode at room temperature and −10 °C, respectively. (**b**,**d**) Shows γ-phase prompter additive in Ni(OH)_2_ cathode at room temperature and −10 °C, respectively [20].

**Figure 3 nanomaterials-10-01878-f003:**
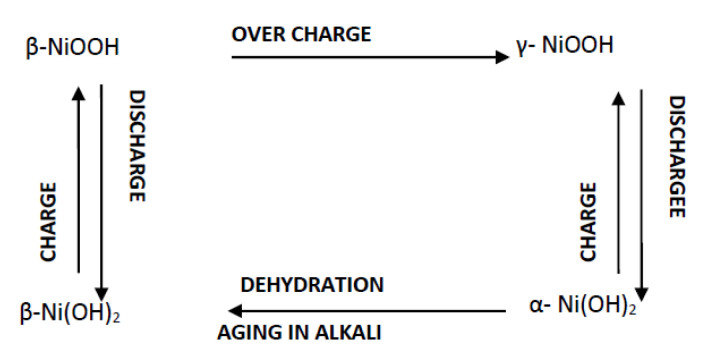
Charge-discharge processes that occur at a nickel hydroxide electrode in a battery system. Reproduced with permission from [29]; Elsevier, 2020.

**Figure 4 nanomaterials-10-01878-f004:**
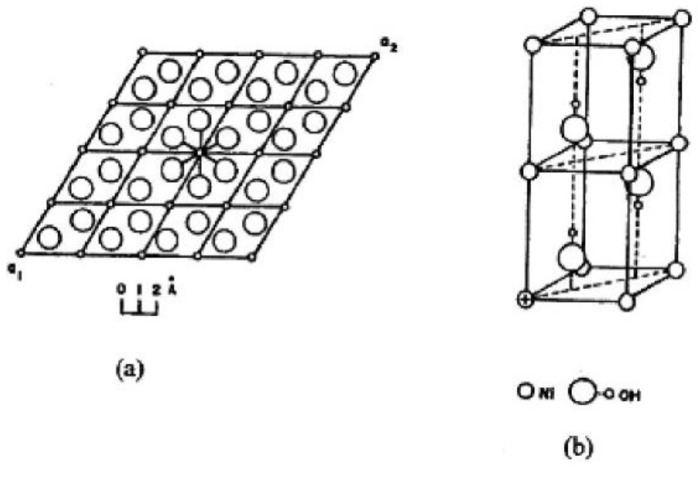
The brucite structure of Ni(OH)_2_: (**a**) hexagonal brucite structure, (**b**) planes stacking indicates the orientation of the O-H bonds. Reproduced with permission from [9]; Elsevier, 2020.

**Figure 5 nanomaterials-10-01878-f005:**
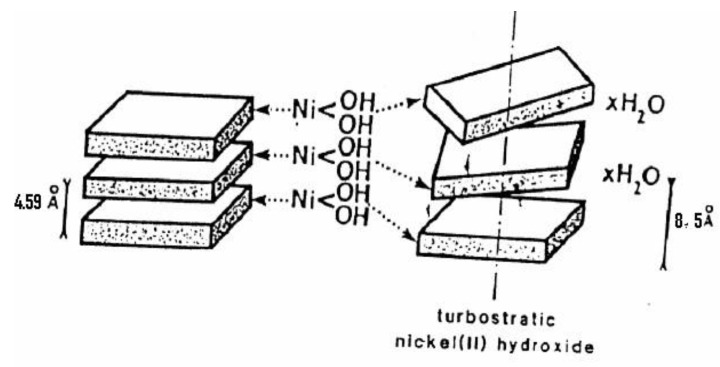
The brucite structure and turbostratic structure of nickel hydroxide showing intercalation with H_2_O molecules.

**Figure 6 nanomaterials-10-01878-f006:**
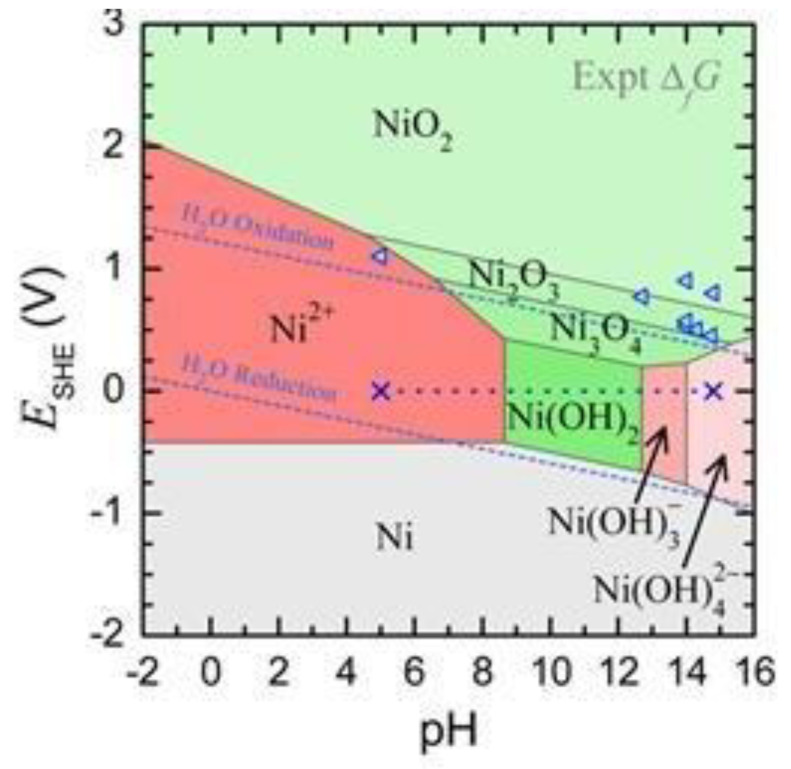
The Pourbaix diagram of nickel hydroxide, showing pH-EMF (V) vs. standard hydrogen electrode (SHE). Reproduced with permission from [39]; American Chemical Society, 2020.

**Figure 7 nanomaterials-10-01878-f007:**
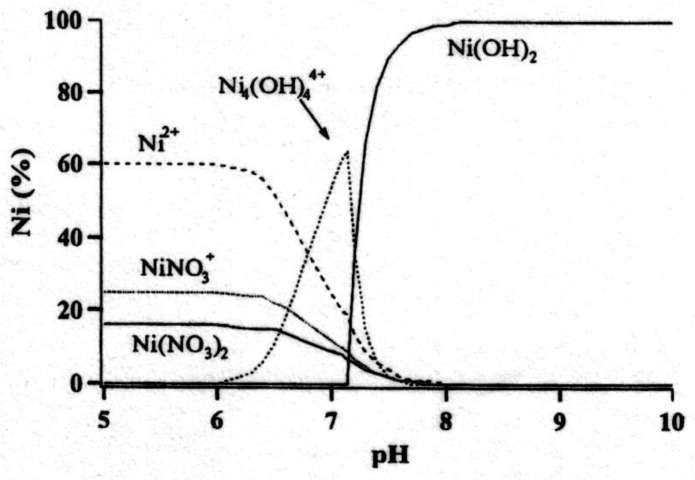
Calculations at equilibrium for the concentration of Ni^++^ species as a function of pH in Ni(NO_3_)_2_ system.

**Figure 8 nanomaterials-10-01878-f008:**
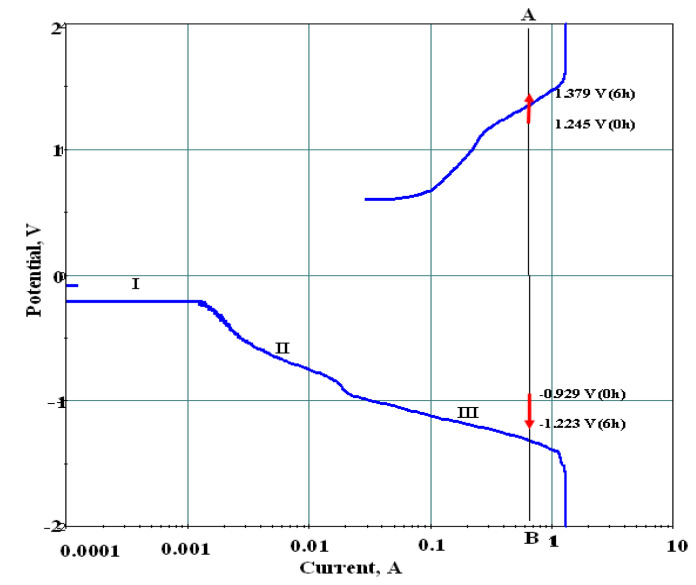
Potentiostatic polarization plots for the electrochemical synthesis of nickel hydroxide employing stainless steel cathode and iridium coated titanium anode. (Scan rate: 50 mV/s, cathode area: 32.5 cm^2^, anode area: 37.5 cm^2^). The letters A and B denote the applied current density at 200 A/m^2^.

**Figure 9 nanomaterials-10-01878-f009:**
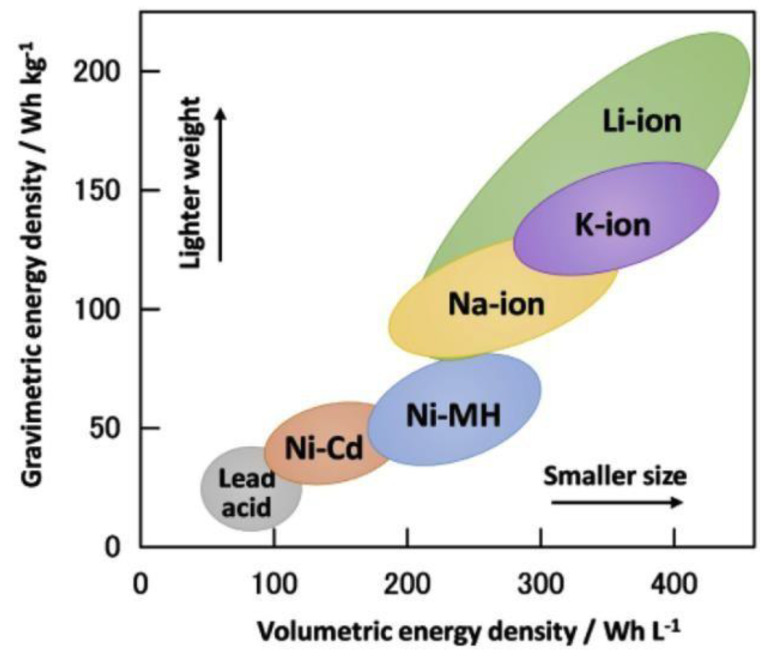
Energy density comparison of commercial battery technologies in the past (lead-acid, Ni-Cd), present (Ni-MH, Li-ion), and the future (Na-ion, and K-ion). Reproduced with permission from [66]; Elsevier, 2020.

**Table 1 nanomaterials-10-01878-t001:** Common commercial battery characteristics compared with aqueous (aq.) Ni-MH system [3,4,5].

Rechargeable Battery System	Electrolyte	Specific Energy (Wh/kg)	Cycle Life	Nominal Voltage (V)
Lead Acid	aq. H_2_SO_4_	30–50	200–300	2.0
Ni-Cd	aq. KOH	45–80	1000	1.2
Ni-MH	aq. KOH	60–120	300–500	1.2
Li-ion (LiCoO_2_)	LiPF_6_ in propylene carbonate/diethyl carbonate	150–250	500–1000	3.6
Li-ion (LiFePO_4_)	LiPF_6_ in ethylene carbonate	90–120	1000–2000	3.3
Li-ion (LiMnO_2_)	LiClO_4_ in ethylene carbonate	100–150	500–1000	3.7

**Table 2 nanomaterials-10-01878-t002:** Representative values from the literature for the precipitation of β*-*nickel hydroxide materials having various Ni structures, prepared at different pH values.

pH	Electrochemically Formed Ni Structure
2.9	Ni^2+^
4.9	Ni^2+^
5.4	Ni^2+^
8.4	Ni(OH)_2_ + NiO + Ni^2+^
14	Ni(OH)^3−^

**Table 3 nanomaterials-10-01878-t003:** Properties of Ni(OH)_2_ prepared at OH^−^/Ni^++^ ratio value of above six.

Sl. No.	Experimental Condition		(OH^−^)/(Ni^++^)	% Ni	No. of Water Content (x)	^#^ TD, g/cc	d50, µm	Surface Area, m^2^/g	^@^ DC, mAh/g
	(Ni^++^), M	* CD, A/m^2^							
1	1.02	500	6.60	58.0	0.48	1.19	44.38	67	120
2	0.34	200	6.72	54.5	0.83	1.15	34.85	110	125

^#^ TD = tap density, * CD = cathode current density, ^@^ DC = discharge capacity.

**Table 4 nanomaterials-10-01878-t004:** Properties of Ni(OH)_2_ prepared at lower OH^−^/Ni^++^ ratios.

Sl. No.	Experimental Condition		(OH^–^)/(Ni^++^)	% Ni	No. of Water Content (x)	^#^ TD, g/cc	d50, µm	Surface Area, m^2^/g	^@^ DC, mAh/g
	(Ni^++^), M	* CD, A/m^2^							
1	1.02	50	0.45	50.5	1.30	1.18	10.56	238	200
2	1.19	200	2.34	52.0	1.12	1.46	19.43	151	140

^#^ TD = tap density, * CD = cathode current density, ^@^ DC = discharge capacity.

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
