# Peer review of "Perspectives on Nickel Hydroxide Electrodes Suitable for Rechargeable Batteries: Electrolytic vs. Chemical Synthesis Routes"

_nanomaterials, 2020, doi:10.3390/nano10091878_

Round 1

Reviewer 1 Report

This manuscript by Asha et al summarizes the synthesis of nickel hydroxide for rechargeable battery application.Chemical ad electrochemical methodes are presented and compared to each other. The electrochemical one shows better control toward the structure of Ni(OH)2 products since the concentration of hydroxide anions can be easily tuned by applied current densities. I think this perspective is very useful and qualified for the publicaton in Nanomaterials.

One additional comment is that the authors may discuss a little bit its role in other electrochemical applications, like oxygen evolution reaction that also needs Ni(OH)2 catalysts. I believe this would expand the readship of this work.

Author Response

This manuscript by Asha et al summarizes the synthesis of nickel hydroxide for rechargeable battery application. Chemical ad electrochemical methods are presented and compared to each other. The electrochemical one shows better control toward the structure of Ni(OH)2 products since the concentration of hydroxide anions can be easily tuned by applied current densities. I think this perspective is very useful and qualified for the publication in Nanomaterials.

One additional comment is that the authors may discuss a little bit its role in other electrochemical applications, like oxygen evolution reaction that also needs Ni(OH)2 catalysts. I believe this would expand the readership of this work.

Response: Many thanks for taking the time to consider our manuscript and making possible these very constructive comments. Please find below a point by point response to the reviewer’s comments.

In the revised version of the manuscript, we have added a section 2.3.3 “Ni-based electrocatalyst for oxygen evolution reaction” As shown below; changes are highlighted in the revised manuscript.

2.3.3 Ni-based electrocatalyst for oxygen evolution reaction

Energy storage will be a critically important part of the transition to a low-emissions future. Currently, there is a heavy reliance on fossil-fuelled energy for both baseload and as a backup for times when renewable output decreases suddenly. Hence, developing clean and sustainable energies and decarbonization are vital for our planet. Clean and green energy such as hydrogen generation using renewable energy resources such as wind and solar as inputs to split water electrochemically into hydrogen and oxygen is potentially attractive on a commercial scale. However, the associated oxygen evolution reaction (OER) from the anode electrode is the efficiency limiting process. Platinum group materials are widely known to be used as OER catalyst, but it is expensive. Therefore, developing cost-effective, highly active, and stable catalysts for OER is crucial. Recently, Sun et al [34] reported the progress on nickel-based oxide and oxyhydroxide composites for water oxidation catalysis through materials design/synthesis and electrochemical performance. The modifications made on these materials are reported in this review article to enhance the oxygen evolution reaction (OER) mechanism and their nanoarchitectures exhibited highly efficient catalysts. The future research trends and perspectives on the development of nickel-based oxygen evolution reaction electrocatalysts are discussed further by Boetcher’s group [35]. They have studied the incorporation of Fe on the improvement of the catalytic activity of nickel hydroxide, which is correlated to the local structure. Apart from nickel-based catalysts, nickel-based compounds have also been reported as the most promising earth-abundant OER catalysts attracting ever-increasing interest due to high activity and stability [36]. In this work, molybdenum (Mo) and iron (Fe) modification appear to have a synergistic effect to enhance both the OER activity and stability with no degradation after 50 h and shown to have to outperform all OER catalyst reported. Overall, nickel oxide as non-noble metal OER has shown potential and in favor of the OER process.

Reviewer 2 Report

The manuscript "Perspectives on Nickel Hydroxide Electrodes Suitable for Rechargeable Batteries: Electrolytic vs. Chemical Synthesis Rout " is very interesting and is well written. The abstract gives a concise summary of the manuscript. The results are also adequate and well analysed/evaluated. The conclusions highlighted and summarised the contents of the manuscript. The manuscripts brings a new perspective Nickel Hydroxide Electrodes for rechargeable batteries. The manuscript will fit really well within the scope of the magazine, therefore, I will recommend its acceptance as it is.

Author Response

The manuscript "Perspectives on Nickel Hydroxide Electrodes Suitable for Rechargeable Batteries: Electrolytic vs. Chemical Synthesis Rout " is very interesting and is well written. The abstract gives a concise summary of the manuscript. The results are also adequate and well analysed/evaluated. The conclusions highlighted and summarised the contents of the manuscript. The manuscripts bring a new perspective Nickel Hydroxide Electrodes for rechargeable batteries. The manuscript will fit really well within the scope of the magazine, therefore, I will recommend its acceptance as it is.

Response: Many thanks for taking the time to consider our manuscript and making possible these very constructive comments.

Reviewer 3 Report

Perspectives on Nickel Hydroxide Electrodes 2 Suitable for Rechargeable Batteries: Electrolytic vs. Chemical Synthesis Routes

Comments: This perspective paper which describes the synthesis methods of nickel hydroxide for nickel-metal hydride batteries is quite impressive, but fails to attract the reader's interest. More review papers are available with a detail study on synthesis methods on “Nickel Hydroxide”. Yet, the following comments are provided to enhance the paper and it can be accepted after it is modified.

  • In line 25-26, it is stated that although thin films have been prepared by the electrochemical route but no significant work has been reported for the synthesis of Ni(OH)2. Check these reports: doi.org/10.1007/s10854-020-03485-6, org/10.1007/s10854-020-03485-6. doi.org/10.1039/C3TA00024A, doi.org/10.1039/C9TC06354D
  • Abstract should be modified.
  • Introduction part lacks the more description of nickel-metal hydride batteries and it compared with other battery systems
  • A table could be provided depending on pH which influences on the structure formation
  • A comparison table can be provided determining the energy density capacity difference between nickel-metal hydride batteries and other batteries for nickel hydroxide.

Reviewer 4 Report

In this work, the authors show the perspectives on nickel hydroxide electrodes for rechargeable batteries. Authors compared through electrolytic and chemical synthesis methods with various electrochemical properties. In my own opinion this paper appears to be acceptable in Nanomaterials. I have several major comments for the authors to improve during the revision:

  1. In the description of basic materials of nickel metal hydride batteries, authors should provide more probable explanation for the low ionic conductivity of metal hydride with low hydrogen holding efficiency. Please adopt or refer the additional references to complete the logic flow.
  2. L84, please make sure the wrong spell. Is it Mi-MH battery or Ni-MH battery?
  3. Section 2.1.1 to 2.1.3 have short information and no previous study, authors should consider to add specific references for each technology.
  4. L319, please make sure the average size unit. Is it um or nm?
  5. L339-340, please make sure the dropping speed or solution unit.      

Round 2

Reviewer 3 Report

The authors have addressed the points in my review.

Reviewer 4 Report

The manuscript has been well modified. The revised manuscript can be acceptable in Nanomaterials.